# Approximation algorithms for stochastic clustering[*]

**David G. Harris**
Department of Computer Science
University of Maryland, College Park, MD 20742
davidgharris29@gmail.com

**Shi Li**
University at Buffalo
Buffalo, NY. shil@buffalo.edu

**Thomas Pensyl**
Bandwidth, Inc.
Raleigh, NC
tpensyl@bandwidth.com

**Aravind Srinivasan**
Department of Computer Science and Institute for Advanced Computer Studies
University of Maryland, College Park, MD 20742
srin@cs.umd.edu

**Khoa Trinh**
Google
Mountain View, CA 94043
khoatrinh@google.com

## Abstract

We consider stochastic settings for clustering, and develop provably-good (approximation) algorithms for a number of these notions. These algorithms allow one to obtain better approximation ratios compared to the usual deterministic clustering setting. Additionally, they offer a number of advantages including providing fairer clustering and clustering which has better long-term behavior for each user. In particular, they ensure that *every user* is guaranteed to get good service (on average). We also complement some of these with impossibility results.
KEYWORDS: clustering, $k$-center, $k$-median, lottery, approximation algorithms

## 1 Introduction

*Clustering* is a fundamental problem in machine learning and data science. A general clustering task is to partition the given data points into clusters such that the points inside the same cluster are "similar" to each other. More formally, consider a set of datapoints $\mathcal{C}$ and a set of "potential cluster centers" $\mathcal{F}$, with a metric $d$ on $\mathcal{C} \cup \mathcal{F}$. We define $n := |\mathcal{C} \cup \mathcal{F}|$. Given any set $\mathcal{S} \subseteq \mathcal{F}$, each $j \in \mathcal{C}$ is associated with the key statistic $d(j, \mathcal{S}) = \min_{i \in S} d(i, j)$. The typical task in a clustering problem is to select a set $\mathcal{S} \subseteq \mathcal{F}$, with a small size, in order to minimize the values of $d(j, \mathcal{S})$. The size of the set $\mathcal{S}$ is often fixed to a value $k$, and we typically "boil down" the large collection of values $d(j, \mathcal{S})$ into a single overall objective function. There are different clustering problems depending on the choice of the objective function and the assumptions on sets $\mathcal{C}$ and $\mathcal{F}$. The most popular problems include

- the $k$-center problem: minimize the maximum value $\max_{j \in \mathcal{C}} d(j, \mathcal{S})$ given that $\mathcal{F} = \mathcal{C}$.

---
[*]Research supported in part by NSF Awards CNS-1010789, CCF-1422569 and CCF-1749864, CCF-1566356, CCF-1717134 and by research awards from Adobe, Inc.

- the $k$-supplier problem: minimize the maximum value $\max_{j \in \mathcal{C}} d(j, \mathcal{S})$ (where $\mathcal{F}$ and $\mathcal{C}$ may be unrelated);

- the $k$-median problem: minimize the summed value $\sum_{j \in \mathcal{C}} d(j, \mathcal{S})$; and

- the $k$-means problem: minimize the summed square value $\sum_{j \in \mathcal{C}} d(j, \mathcal{S})^2$.

An important special case is when $\mathcal{C} = \mathcal{F}$ (e.g. the $k$-center problem); since this often occurs in the context of data clustering, we refer to this case as the **self-contained clustering (SCC)** setting. In the general case, $\mathcal{C}$ and $\mathcal{F}$ may be unrelated (intersect with each other arbitrarily).

These classic NP-hard problems have been studied intensively for the past few decades. There is an alternative interpretation of these clustering problems from the viewpoint of operations research: the sets $\mathcal{F}$ and $\mathcal{C}$ can be thought of as "facilities" and "clients", respectively. We say that a $i \in \mathcal{F}$ is *open* if $i$ is placed into the solution set $\mathcal{S}$. For a set $\mathcal{S} \subseteq \mathcal{F}$ of open facilities, $d(j, \mathcal{S})$ can then be interpreted as the connection cost of client $j$. The above-mentioned problems try to optimize the cost of serving all clients in different ways. This terminology has historically been used for $k$-center type clustering problems, and so will adopt this throughout for consistency here. However, our focus is on the case in which $\mathcal{C}$ and $\mathcal{F}$ are arbitrary given sets in the data-clustering setting.

Since these problems are NP-hard, much effort has been paid on algorithms with "small" provable *approximation ratios/guarantees*: i.e., polynomial-time algorithms that produce solutions of cost at most $\alpha$ times the optimal. The current best-known approximation ratio for $k$-median is $2.675$ by Byrka et. al. [6] and it is NP-hard to approximate this problem to within a factor of $1 + 2/e \approx 1.735$ [15]. The recent breakthrough by Ahmadian et. al. [1] gives a $6.357$-approximation algorithm for $k$-means, improving on the previous approximation guarantee of $(9 + \epsilon)$ based on local search [16]. Finally, the $k$-supplier problem is "easier" than both $k$-median and $k$-means in the sense that a simple 3-approximation algorithm [12] is known, as is a 2-approximation for $k$-center problem: we cannot do better than these approximation ratios unless P = NP [12].

While optimal approximation algorithms for the center-type problems are well-known, all of the current algorithms give deterministic solutions. One can easily demonstrate instances where such algorithms return a worst-possible solution: (i) all clusters have the same worst-possible radius ($2T$ for $k$-center and $3T$ for $k$-supplier where $T$ is the optimal radius) and (ii) almost all data points are on the circumference of the resulting clusters. Although it is NP-hard to improve the approximation ratios here, our new randomized algorithms provide significantly-improved "per-point" guarantees. For example, we achieve a new "per-point" guarantee $\mathbf{E}[d(j, \mathcal{S})] \leq (1 + 2/e)T \approx 1.736T$, while respecting the usual guarantee $d(j, \mathcal{S}) \leq 3T$ with probability one. *Thus, while maintaining good global quality with probability one, we also provide superior stochastic guarantees for each user.*

In this paper, we study generalized variants of the center-type problems where $\mathcal{S}$ is drawn from a probability distribution over $\binom{\mathcal{F}}{k}$ (where $\binom{\mathcal{F}}{k}$ denotes the set of $k$-element subsets of $\mathcal{F}$); we refer to these as $k$-*lotteries.* We aim to construct a $k$-lottery $\Omega$, which achieves certain guarantees on the distributional properties or expected value of $d(j, \mathcal{S})$. The $k$-center problem can be viewed as the special case in which the distribution $\Omega$ in supported on a single point (we refer to this situation by saying that $\Omega$ is *deterministic*). Our goal is to find an *approximating distribution* $\tilde{\Omega}$ which matches the *target distribution* $\Omega$ as closely as possible for each client $j$.

We have seen that stochastic solutions allows one to go beyond the approximation hardness of a number of classical facility location problems. In addition to allowing higher-quality solutions, there are a number of applications where clustering based on expected distances can be beneficial. We summarize three here: smoothing the integrality constraints of clustering, solving repeated problem instances, and achieving fair solutions.

**Stochasticity as interpolation.** For many problems in practice, *robustness* of the solution is often more important than achieving the absolute optimal value for the objective function. Our stochastic perspective is very useful here. One potential problem with the center measure is that it can be highly non-robust; adding a single new point may drastically change the overall objective function. As an extreme example, consider $k$-center with $k$ points, each at distance 1 from each other. This clearly has value 0 (choosing $\mathcal{S} = \mathcal{C}$). However, if a single new point at distance 1 to all other points is added, then the solution jumps to 1. By choosing $k$ facilities uniformly at random among the full set of $k + 1$, we can ensure that $\mathbf{E}[d(j, \mathcal{S})] = \frac{1}{k+1}$ for every point $j$, a much smoother transition.

**Repeated clustering problems.** Consider clustering problems where the choice of $\mathcal{S}$ can be changed periodically: e.g., $\mathcal{S}$ could be the set of $k$ locations in the cloud where a service-provider chooses to provide services from. This set $\mathcal{S}$ can be shuffled periodically in a manner transparent to end-users. For any user $j \in \mathcal{C}$, the statistic $d(j, \mathcal{S})$ represents the latency of the service $j$ receives (from its closest service-point in $\mathcal{S}$). If we aim for a fair or minmax service allocation, then our $k$-center stochastic approximation results ensure that for *every client $j$*, the long-term average service-time – where the average is taken over the periodic re-provisioning of $\mathcal{S}$ – is at most around $1.736T$ with high probability. (Furthermore, we have the risk-avoidance guarantee that in no individual provisioning of $\mathcal{S}$ will any client have service-time greater than $3T$.) We emphasize that this type of guarantee pertains to multi-round clustering problems, and is *not* by itself stochastic. This is difficult to achieve for the usual type of approximation algorithms and impossible for "stateless" deterministic algorithms.

**Fairness in clustering.** The classical clustering problems combine the needs of many different points (elements of $\mathcal{C}$) into one metric. However, clustering (and indeed many other ML problems) are increasingly driven by inputs from parties with diverse interests. Fairness in these contexts is a challenging issue to address; this concern has taken on greater importance in our current world, where decisions will increasingly be taken by algorithms and machine learning. Representative examples of recent concerns include the accusations of, and fixes for, possible racial bias in Airbnb rentals [4] and the finding that setting the gender to "female" in Google's *Ad Settings* resulted in getting fewer ads for high-paying jobs [8]. Starting with older work such as [21], there have been highly-publicized works on bias in allocating scarce resources – e.g., racial discrimination in hiring applicants who have very similar resumés [5]. Additional work discusses the possibility of bias in electronic marketplaces, whether human-mediated or not [3, 4].

In such settings, a fair allocation should provide good service guarantees *to each user individually*. In data clustering settings where a user corresponds to a datapoint, this means that every point $j \in \mathcal{C}$ should be guaranteed a good value of $d(j, \mathcal{S})$. This is essentially what $k$-center type problems are aiming for, but the stochastic setting broadens the meaning of good per-user service.

Consider the following scenarios. Each user submits their data (corresponding to a point in $\mathcal{C}$) – as is increasingly common, explicitly or implicitly – to an aggregator such as an e-commerce or other site. The cluster centers are "influencer" nodes that the aggregator tries to connect users with in some way. Two examples that motivate the aggregator's budget on $k$ are: (i) the aggregator can give a free product sample to each influencer to influence the whole population in aggregate, as in [17], and (ii) in a resource-constrained setting, the aggregator forms a sparse "sketch" with $k$ nodes (the cluster centers), with each cluster center hopefully being similar to the nodes in its cluster so that the nodes get relevant recommendations (in a recommender-like system). Each point $j$ would like to be in a cluster that is "high quality" *from its perspective*, with $d(j, \mathcal{S})$ being a good proxy for such quality. Indeed, there is increasing emphasis on the fact that organizations monetize their user data, and that users need to be compensated for this (see, e.g., the well-known works of Lanier and others [19, 14]). This is a transition from viewing data as capital to viewing *data as labor*. A concrete way for users (i.e., the data points $j \in \mathcal{C}$) to be compensated in our context is for each user to get a guarantee on their solution quality: i.e., bounds on $d(j, \mathcal{S})$.

## 1.1 Our contributions and overview

In Section 2, we encounter the first type of approximation algorithm which we refer to as *chance $k$-coverage*: namely, where every client $j$ has a distance demand $r_j$ and probability demand $p_j$, and we wish to find a distribution satisfying $\Pr[d(j, \mathcal{S}) \leq r_j] \geq p_j$. We show how to obtain an approximation algorithm to find an approximating distribution $\tilde{\Omega}$ with[2]

$$\Pr_{\mathcal{S} \sim \tilde{\Omega}}[d(j, \mathcal{S}) \leq 9r_j] \geq p_j.$$

In a number of special cases, such as when all the values of $p_j$ or $r_j$ are the same, the distance factor 9 can be improved to 3, *which is optimal*; it is an interesting question whether we can improve the coefficient "9" to its best-possible value in the general case.

In Section 3, we consider a special case of chance $k$-coverage, in which $p_j = 1$ for all clients $j$. This is equivalent to the classical (deterministic) $k$-supplier problem. By allowing the approximating

distribution $\tilde{\Omega}$ to be stochastic, we are able to achieve significantly better distance guarantees than are possible for $k$-supplier. For instance, we are able to approximate the $k$-center problem by finding an approximating distribution $\tilde{\Omega}$ with

$$\mathbf{E}_{\mathcal{S}\sim\tilde{\Omega}}[d(j,\mathcal{S})] \leq 1.592T \text{ and } \Pr[d(j,\mathcal{S}) \leq 3T] = 1$$

where $T$ is the optimal solution to the (deterministic) $k$-center problem. By contrast, deterministic polynomial-time algorithms cannot guarantee $d(j,\mathcal{S}) < 2T$ for all $j$, unless P = NP [12].

In Section 4, we show a variety of lower bounds on the approximation factors achievable by efficient algorithms (assuming that $P \neq NP$). For instance, we show that our approximation algorithm for homogeneous chance $k$-coverage has the optimal distance approximation factor 3, that our approximation algorithm for $k$-supplier has optimal approximation factor $1 + 2/e$, and that the approximation factor 1.592 for $k$-center cannot be improved below $1 + 1/e$.

## 1.2 Related work

While most analysis for facility location problems has focused on the static case, there have been other works analyzing a similar lottery model for center-type problems. In [11, 10], Harris et. al. analyzed models similar to chance $k$-coverage and minimization of $\mathbf{E}[d(j,\mathcal{S})]$, but applied to knapsack center and matroid center problems; they also considered robust versions (in which a small subset of clients may be denied service). While the overall model was similar to the ones we explore here, the techniques are somewhat different. In particular, the works [11, 10] focus on *approximately* satisfying the knapsack constraints; this is very different from the problem of opening exactly $k$ cluster centers, which we mostly cover here.

Similar types of stochastic approximation guarantees have appeared in the context of developing approximation algorithms for static problems, particularly $k$-median problems. In [7], Charikar & Li discussed a randomized procedure for converting a linear-programming relaxation in which a client has *fractional* distance $t_j$, into a distribution $\Omega$ satisfying $\mathbf{E}_{\mathcal{S}\sim\Omega}[d(j,\mathcal{S})] \leq 3.25t_j$. This property can be used, among other things, to achieve a 3.25-approximation for $k$-median. However, many other randomized rounding algorithms for $k$-median only seek to preserve the *aggregate* value $\sum_j \mathbf{E}[d(j,\mathcal{S})]$, without our type of per-point guarantee.

We also contrast our approach with a different stochastic $k$-center problem considered in works such as [13, 2]. These consider a model with a fixed, deterministic set $\mathcal{S}$ of open facilities, while the client set is determined stochastically; this model is almost precisely opposite to ours.

## 1.3 Notation

We will let $[t]$ denote the set $\{1, 2, \ldots, t\}$. We use the Iverson notation throughout, so that for any Boolean predicate $\mathcal{P}$ we let $[[\mathcal{P}]]$ be equal to one if $\mathcal{P}$ is true and zero otherwise. For any vector $a = (a_1, \ldots, a_n)$ and a subset $X \subseteq [n]$, we often write $a(X)$ as shorthand for $\sum_{i \in X} a_i$. Given any $j \in \mathcal{C}$ and any real number $r \geq 0$, we define the ball $B(j, r) = \{i \in \mathcal{F} \mid d(i, j) \leq r\}$. We let $V_j$ denote the facility closest to $j$. Note that $V_j = j$ in the SCC setting.

## 1.4 Some useful subroutines

There are two basic subroutines that will be used repeatedly in our algorithms: *dependent rounding* and *greedy clustering*.

**Proposition 1.1** ([22]). *There exists a randomized polynomial-time algorithm* DEPROUND$(y)$ *which takes as input a vector* $y \in [0, 1]^n$, *and in polynomial time outputs a random set* $Y \subseteq [n]$ *with the following properties:*

*(P1)* $\Pr[i \in Y] = y_i$, *for all* $i \in [n]$,

*(P2)* $\lfloor \sum_{i=1}^{n} y_i \rfloor \leq |Y| \leq \lceil \sum_{i=1}^{n} y_i \rceil$ *with probability one,*

*(P3)* $\Pr[Y \cap S = \emptyset] \leq \prod_{i \in S}(1 - y_i)$ *for all* $S \subseteq [n]$.

We adopt the following additional convention: suppose $(y_1, \ldots, y_n) \in [0, 1]^n$ and $S \subseteq [n]$; we then define DEPROUND$(y, S) \subseteq S$ to be DEPROUND$(x)$, for the vector $x$ defined by $x_i = y_i[[i \in S]]$.

The greedy clustering procedure takes an input a set of weights $w_j$ and sets $F_j \subseteq \mathcal{F}$ for every client $j \in \mathcal{C}$, and executes the following procedure:

---

**Algorithm 1** GREEDYCLUSTER$(F, w)$

---
1: Sort $\mathcal{C}$ as $\mathcal{C} = \{j_1, j_2, \dots, j_\ell\}$ where $w_{j_1} \leq w_{j_2} \leq \cdots \leq w_{j_\ell}$.
2: Initialize $C' \leftarrow \emptyset$
3: **for** $t = 1, \dots, \ell$ **do**
4:     **if** $F_{j_t} \cap F_{j'} = \emptyset$ for all $j' \in C'$ **then** update $C' \leftarrow C' \cup \{j_t\}$
5: Return $C'$

---

**Observation 1.2.** *If $C' = $ GREEDYCLUSTER$(F, w)$ then for any $j \in \mathcal{C}$ there is $z \in C'$ with $w_z \leq w_j$ and $F_z \cap F_j \neq \emptyset$.*

## 2 The chance $k$-coverage Problem

In this section, we consider a scenario we refer to as the *chance k-coverage problem*, wherein every point $j \in \mathcal{C}$ has demand parameters $p_j, r_j$, and we wish to find a $k$-lottery $\Omega$ such that

$$\Pr_{\mathcal{S} \sim \Omega}[d(j, \mathcal{S}) \leq r_j] \geq p_j. \tag{1}$$

If a set of demand parameters has a $k$-lottery satisfying (1), we say that they are *feasible*. We refer to the special cases wherein every client $j$ has a common value $p_j = p$, or every client $j$ has a common value $r_j = r$ (or both), as *homogeneous*. This often arises in the context of fair allocations, for example, $k$-supplier is a special case of the homogeneous chance $k$-coverage problem, in which $p_j = 1$ and $r_j$ is equal to the optimal $k$-supplier radius.

As we discuss in Section 4, any approximation algorithm must either significantly give up a guarantee on the distance, or probability (or both). Our approximation algorithms for chance $k$-coverage will be based on the following linear programming (LP) relaxation, which we refer to as the *chance LP*. We consider the following polytope $\mathcal{P}$ over variables $b_i$, where $i$ ranges over $\mathcal{F}$ ($b_i$ represents the probability of opening facility $i$):

(B1) $\sum_{i \in B(j, r_j)} b_i \geq p_j$ for all $j \in \mathcal{C}$,

(B2) $b(\mathcal{F}) = k$,

(B3) $b_i \in [0, 1]$ for all $i \in \mathcal{F}$.

For the remainder of this section, we assume we have found a vector $b \in \mathcal{P}$. By a standard center-splitting step, we also generate, for every point $j \in C$, a center set $F_j \subseteq B(j, r_j)$ with $b(F_j) = p_j$.

**Theorem 2.1.** *If $p, r$ is feasible then one may efficiently construct a $k$-lottery $\Omega$ satisfying $\Pr_{\mathcal{S} \sim \Omega}[d(j, \mathcal{S}) \leq r_j] \geq (1 - 1/e)p_j$.*

### 2.1 Distance approximation for chance $k$-coverage

We next will show how to satisfy the probability constraint exactly, with a constant-factor loss in the distance guarantee. This algorithm will be based on iterated randomized rounding of the vector $b$. This is similar to a method of [18], which also uses iterative rounding for a (deterministic) $k$-median and $k$-means approximations.

We will maintain a vector $b \in [0, 1]^{\mathcal{F}}$ and maintain two sets of points $C_{\text{tight}}$ and $C_{\text{slack}}$, with the following properties:

(C1) $C_{\text{tight}} \cap C_{\text{slack}} = \emptyset$.

(C2) For all $j, j' \in C_{\text{tight}}$, we have $F_j \cap F_{j'} = \emptyset$

(C3) Every $j \in C_{\text{tight}}$ has $b(F_j) = 1$,

(C4) Every $j \in C_{\text{slack}}$ has $b(F_j) \leq 1$.

(C5) We have $b(\bigcup_{j \in C_{\text{tight}} \cup C_{\text{slack}}} F_j) \leq k$

Given our initial LP solution $b$, setting $C_{\text{tight}} = \emptyset, C_{\text{slack}} = \mathcal{C}$ will satisfy criteria (C1)–(C5); note that (C4) holds as $b(F_j) = p_j \leq 1$ for all $j \in \mathcal{C}$.

**Proposition 2.2.** *Given any vector $b \in [0,1]^{\mathcal{F}}$ satisfying constraints (C1)—(C5) with $C_{slack} \neq \emptyset$, it is possible to generate a random vector $b' \in [0,1]^{\mathcal{F}}$ such that $\mathbf{E}[b'] = b$, and with probability one $b'$ satisfies constraints (C1) — (C5) as well as having some $j \in C_{slack}$ with $b'(F_j) \in \{0,1\}$.*

We can now describe our iterative rounding algorithm, Algorithm 2.

---

**Algorithm 2** Iterative rounding algorithm for chance $k$-coverage

---

1: Let $b$ be a fractional LP solution and form the corresponding sets $F_j$.
2: Initialize $C_{\text{tight}} = \emptyset, C_{\text{slack}} = \mathcal{C}$
3: **while** $C_{\text{slack}} \neq \emptyset$ **do**
4:    Draw a fractional solution $b'$ with $\mathbf{E}[b'] = b$ according to Proposition 2.2.
5:    Select some $w \in C_{\text{slack}}$ with $b'(F_w) \in \{0,1\}$.
6:    Update $C_{\text{slack}} \leftarrow C_{\text{slack}} - \{w\}$
7:    **if** $b'(F_w) = 1$ **then**
8:       Update $C_{\text{tight}} \leftarrow C_{\text{tight}} \cup \{w\}$
9:       **if** there is any $z \in C_{\text{tight}} \cup C_{\text{slack}}$ such that $r_z \geq r_w/2$ and $F_z \cap F_w \neq \emptyset$ **then**
10:          Update $C_{\text{tight}} \leftarrow C_{\text{tight}} - \{z\}, C_{\text{slack}} \leftarrow C_{\text{slack}} - \{z\}$
11:    Update $b \leftarrow b'$.
12: For each $j \in C_{\text{tight}}$, open an arbitrary center in $F_j$.

---

**Theorem 2.3.** *Every $j \in \mathcal{C}$ has $\Pr[d(j, \mathcal{S}) \leq 9r_j] \geq p_j$.*

## 2.2 Approximation algorithm for uniform $p$ or $r$

The distance guarantee can be significantly improved in two natural cases: when all the values of $p_j$ are the same, or when all the values of $r_j$ are the same.

We use a similar algorithm for both these cases. The main idea is to first select a set $C'$ according to some greedy order, such that the cluster sets $\{F_{j'} \mid j' \in C'\}$ intersect all the cluster $F_j$. We then open a single item from each cluster. The only difference between the two cases is the choice of weighting function $w_j$ for the greedy cluster selection. We can summarize these algorithms as follows:

---

**Algorithm 3** Rounding algorithm for $k$-coverage for uniform $p$ or uniform $r$.

---

1: Set $C' = \textsc{GreedyCluster}(F_j, w_j)$
2: Set $Y = \textsc{DepRound}(p, C')$
3: Form solution set $\mathcal{S} = \{V_j \mid j \in Y\}$.

---

**Proposition 2.4.** *Algorithm 3 opens at most $k$ facilities.*

*Proof.* The dependent rounding step ensures that $\sum_{j \in Y} p_j \leq \lceil \sum_{j \in C'} p_j \rceil = \lceil \sum_{j \in C'} b(F_j) \rceil \leq \lceil \sum_{i \in \mathcal{F}} b_i \rceil \leq k$. $\square$

**Proposition 2.5.** *Suppose that $p_j$ is the same for every $j \in \mathcal{C}$. Then using the weighting function $w_j = r_j$ ensures that every $j \in \mathcal{C}$ it satisfies $\Pr[d(j, \mathcal{S}) \leq 3r_j] \geq p_j$. Furthermore, in the SCC setting, it satisfies $\Pr[d(j, \mathcal{S}) \leq 2r_j] \geq p_j$.*

**Proposition 2.6.** *Suppose $r_j$ is the same for every $j \in C$. Then using the weighting function $w_j = 1 - p_j$ ensures that every $j \in \mathcal{C}$ satisfies $\Pr[d(j, \mathcal{S}) \leq 3r_j] \geq p_j$. Furthermore, in the SCC setting, it satisfies $\Pr[d(j, \mathcal{S}) \leq 2r_j] \geq p_j$.*

# 3 Chance $k$-coverage: the deterministic case

An important special case of the $k$-coverage problem is the one where $p_j = 1$ for all $j \in \mathcal{C}$. In this case, the target distribution $\Omega$ is just a single set $\mathcal{S}$ satisfying $d(j, \mathcal{S}) \leq r_j$. In the homogeneous setting (where all the $r_j$ are equal to the same value), this is specifically the $k$-center or $k$-supplier

problem. The typical approach to approximate this is to have the approximation distribution $\tilde{\Omega}$ also be a single set $\mathcal{S}'$; in such a case the best guarantee available is $d(j, \mathcal{S}) \leq 3r_j$.

We improve the distance guarantee by allowing $\tilde{\Omega}$ to be a distribution. Specifically, we construct a $k$-lottery $\tilde{\Omega}$ such that $d(j, \mathcal{S}) \leq 3r_j$ with probability one, and $\mathbf{E}_{\mathcal{S} \sim \tilde{\Omega}}[d(j, \mathcal{S})] \leq cr_j$, where the constant $c$ satisfies the following bounds:

1. In the general case, $c = 1 + 2/e \approx 1.73576$;

2. In the SCC setting, $c = 1.60793$;

3. In the homogeneous SCC setting, $c = 1.592$.

By a reduction to set cover, we will show that the constant value $1 + 2/e$ is optimal for the general case (even when all the $r_j$ are equal), and for the third case the constant $c$ cannot be made lower than $1 + 1/e \approx 1.367$.

We also remark that this type of stochastic guarantee allows us to efficiently construct publicly-verifiable lotteries. This means that the server locations are not only fair, but they are also transparent and seen to be fair.

**Proposition 3.1.** *Let $\epsilon > 0$. Suppose that there is an efficiently-samplable $k$-lottery $\Omega$ with $\Pr_{\mathcal{S} \sim \Omega}[d(j, \mathcal{S}) \leq c_1 r_j] = 1$ and $\mathbf{E}_{\mathcal{S} \sim \Omega}[d(j, \mathcal{S})] \leq c_2 r_j$, for constants $c_2 \leq c_1$. Then there is an expected polynomial time procedure to construct an explicitly-enumerated $k$-lottery $\Omega'$, with support size $|\Omega'| = O(\frac{\log n}{\epsilon^2})$, such that $\Pr_{\mathcal{S} \sim \Omega'}[d(j, \mathcal{S}) \leq c_1 r_j] = 1$ and $\mathbf{E}_{\mathcal{S} \sim \Omega'}[d(j, \mathcal{S})] \leq (c_2 + \epsilon)r_j$.*

### 3.1 Randomized rounding via clusters

We use a similar type of algorithm to that considered in Section 2.2: we choose a covering set of clusters $C'$, and we open exactly one item from each cluster. The main difference is that instead of opening the nearest item $V_j$ for each $j \in C'$, we instead open a cluster according to the LP solution $b_i$.

---

**Algorithm 4** Rounding algorithm for $k$-supplier

1: Set $C' = \text{GREEDYCLUSTER}(F_j, r_j)$.
2: Set $F_0 = \mathcal{F} - \bigcup_{j \in C'} F_j$
3: **for** $j \in C'$ **do**
4:      Randomly select $W_j \in F_j$ according to the distribution $\Pr[W_j = i] = b_i$
5: Let $\mathcal{S}_0 \leftarrow \text{DEPROUND}(b, F_0)$
6: Return $\mathcal{S} = \mathcal{S}_0 \cup \{W_j \mid j \in C'\}$

---

**Theorem 3.2.** *For any $w \in \mathcal{C}$, the solution set $\mathcal{S}$ of Algorithm 4 satisfies $d(w, \mathcal{S}) \leq 3r_w$ with probability one and $\mathbf{E}[d(w, \mathcal{S})] \leq (1 + 2/e)r_w$.*

In the SCC setting, we may improve the approximation ratio using the following Algorithm 5; it is the same as Algorithm 4, except that we have moved some values of $b$ to the cluster centers.

---

**Algorithm 5** Rounding algorithm for $k$-center

1: Set $C' = \text{GREEDYCLUSTER}(F_j, r_j)$.
2: Set $F_0 = \mathcal{F} - \bigcup_{j \in C'} F_j$
3: **for** $j \in C'$ **do**
4:      Randomly select $W_j \in F_j$ according to the distribution $\Pr[W_j = i] = (1 - q)b_i + q[[i = j]]$
5: Let $\mathcal{S}_0 \leftarrow \text{DEPROUND}(b, F_0)$
6: Return $\mathcal{S} = \mathcal{S}_0 \cup \{W_j \mid j \in C'\}$

---

**Theorem 3.3.** *Let $w \in \mathcal{C}$. After running Algorithm 5 with $q = 0.464587$ we have $d(w, \mathcal{S}) \leq 3r_w$ with probability one and $\mathbf{E}[d(w, \mathcal{S})] \leq 1.60793r_w$.*

## 3.2 The homogeneous SCC setting

The SCC setting, in which all the values of $r_j$ are equal, is equivalent to the classical $k$-center problem. We will improve on Theorem 3.3 through a more complicated rounding process based on greedily-chosen partial clusters. Specifically, we select cluster centers $\pi(1), \ldots \pi(n)$, wherein $\pi(i)$ is chosen to maximize $b(F_{\pi(i)} - F_{\pi(1)} - \cdots - F_{\pi(i-1)})$. By renumbering $\mathcal{C}$, we may assume without loss of generality that the resulting permutation $\pi$ is the identity; therefore, we assume throughout this section that $\mathcal{C} = \mathcal{F} = [n]$ and for all $i, j \in [n]$ we have

$$b(F_i - F_1 - \cdots - F_{i-1}) \geq b(F_j - F_1 - \cdots - F_{i-1})$$

For each $j \in [n]$, we let $G_j = F_j - F_1 - \cdots - F_{j-1}$; we refer to $G_j$ as a *cluster* and we let $z_j = b(G_j)$. We say that $G_j$ is a *full cluster* if $z_j = 1$ and a *partial cluster* otherwise.

We use the following randomized rounding strategy to select the centers. We begin by choosing two real numbers $Q_f, Q_p$ (short for *full* and *partial*); these are drawn according to a joint probability distribution which we discuss later. Recall our notational convention that $\bar{q} = 1 - q$; this notation will be used extensively in this section to simplify the formulas.

We then use Algorithm 7:

---
**Algorithm 7** Partial-cluster based algorithm for $k$-center
---
1:   $Z \leftarrow \text{DepRound}(z)$
2: **for** $j \in Z$ **do**
3:     **if** $z_j = 1$ **then**
4:       Randomly select a point $W_j \in G_j$ according to the distribution
$$\Pr[W_j = i] = (1 - Q_f)y_i + Q_f[[i = j]]$$
5:     **else**
6:       Randomly select a point $W_j \in G_j$ according to the distribution
$$\Pr[W_j = i] = ((1 - Q_p)y_i + Q_p[[i = j]])/z_j$$
7: Return $\mathcal{S} = \{W_j \mid j \in Z\}$

---

Let us give some intuitive motivation for Algorithm 7. Consider some $j \in \mathcal{C}$. It may be beneficial to open the center of some cluster near $j$ as this will ensures $d(j, \mathcal{S}) \leq 2$. However, there is no benefit to opening the centers of multiple clusters. So, we would like to ensure that there is a significant negative correlation between opening the centers of distinct clusters near $j$. Unfortunately, there does not seem to be any way to achieve this with respect to full clusters — as all full clusters "look alike," we cannot enforce any significant negative correlation among the indicator random variables for opening their centers. By taking advantage of partial clusters, we are able to break this symmetry.

**Theorem 3.4.** *Suppose that $Q_f, Q_p$ has the following distribution:*

$$(Q_f, Q_p) = \begin{cases} (0.4525, 0) & \text{with prob } p = 0.773 \\ (0.0480, 0.3950) & \text{with prob } 1 - p \end{cases}.$$

*Then for all $i \in \mathcal{F}$ we have $d(i, \mathcal{S}) \leq 3r_i$ with probability one, and $\mathbf{E}[d(i, \mathcal{S})] \leq 1.592 r_i$.*

## 4 Lower bounds on approximation ratios

We next show lower bounds corresponding to optimization problem for chance $k$-coverage analyzed in Sections 2, and 3. These constructions are adapted from similar lower bounds for approximability of $k$-median [9], which in turn are based on the hardness of set cover. In a set cover instance, we have a ground set $[n]$, as well as a collection of sets $\mathcal{B} = \{S_1, \ldots, S_m\} \subseteq 2^{[n]}$. The goal is to select a collection $X \subseteq [m]$ such that the sets $\cup_{i \in X} S_i = [n]$, and such that $|X|$ is minimized. The minimum value of $|X|$ thus obtained is denoted by OPT.

We quote a result of Moshovitz [20] on the inapproximability of set cover.

**Theorem 4.1** ([20])**.** *Assuming $P \neq NP$, there is no algorithm running in polynomial time which guarantees a set-cover solution $X$ with $S_X = [n]$ and $|X| \leq (1 - \epsilon) \ln n \times OPT$, where $\epsilon > 0$ is any constant.*

For the remainder of this section, we assume that $P \neq NP$. We will need a simple corollary of Theorem 4.1, which is a (slight reformulation) of the hardness of approximating max-coverage.

**Corollary 4.2.** *Assuming $P \neq NP$, there is no polynomial-time algorithm which guarantees a set-cover solution $X$ with $|X| \leq OPT$ and $|S_X| \geq cn$ for any constant $c > 1 - 1/e$.*

We can encode facility location problems as special cases of max-coverage. Let us sketch how this works in the non-SCC setting.

Consider a set cover instance $\mathcal{B} = \{S_1, \ldots, S_m\}$ on ground set $[n]$. We begin by guessing the value OPT (there are at most $m$ possibilities, so this can be done in polynomial time). We define a $k$-center instance with $k = $ OPT. We define disjoint client and facility sets, where $\mathcal{F}$ is identified with $[m]$ and $\mathcal{C}$ is identified with $[n]$. The distance is defined by $d(i, j) = 1$ if $j \in S_i$ and $d(i, j) = 3$ otherwise.

Now note that if $X$ is an optimal solution to $\mathcal{B}$ then $d(j, X) \leq 1$ for all points $j \in \mathcal{C}$. So there exists a (deterministic) distribution achieving $r_j = 1$. On the other hand, suppose that $\mathcal{A}$ generates a solution $X \in \binom{\mathcal{F}}{k}$ where $\mathbf{E}[d(j, X)] \leq cr_j$; the set $X$ can also be regarded as a solution to the set cover instance.

Thus, if a polynomial-time algorithm $\mathcal{A}$ generates a distribution $\Omega$ which ensures that $\mathbf{E}[d(j, X)] \leq cr_j$, it also generates a solution a fraction $(3 - c)/2$ of the sets in $\mathcal{B}$. By Corollary 4.2, this means that we must have $(3 - c)/2 \geq 1 - 1/e$, i.e. $c \geq 1 + 2/e$.

The construction for SCC instances and for chance $k$-coverage problems is similar, with a few more technical details.

**Theorem 4.3.** *Assuming $P \neq NP$, no polynomial-time algorithm can guarantee $\mathbf{E}[d(j, \mathcal{S})] \leq cr_j$ for $c < 1 + 2/e$, even if all $r_j$ are equal to each other. Thus, the approximation constant in Theorem 3.2 cannot be improved.*

**Theorem 4.4.** *Assuming $P \neq NP$, no polynomial-time algorithm can guarantee $\mathbf{E}[d(j, \mathcal{S})] \leq cr_j$ for $c < 1 + 1/e$, even if all $r_j$ are equal to each other and $\mathcal{C} = \mathcal{F}$. Thus, the approximation factor 1.592 in Theorem 3.4 cannot be improved below $1 + 1/e$.*

**Proposition 4.5.** *Assuming $P \neq NP$, no polynomial-time algorithm can take as input a feasible vector $r, p$ for the chance $k$-coverage problem and generate a $k$-lottery $\Omega$ guaranteeing either $\Pr_{\mathcal{S} \sim \Omega}[d(j, \mathcal{S}) < r_j] \geq (1 - 1/e - \epsilon)p_j$, or $\Pr_{\mathcal{S} \sim \Omega}[d(j, \mathcal{S}) < 3r_j] \geq p_j$, for any constant $\epsilon > 0$. This holds even when restricted to homogeneous problem instances. Likewise, in the homogeneous SCC setting, we cannot ensure that $\Pr_{\mathcal{S} \sim \Omega}[d(j, \mathcal{S}) < 2r_j] \geq p_j$,*

# 5 Acknowledgments

Thanks to the anonymous conference referees, for many useful suggestions and for helping to tighten the focus of the paper.

## Footnotes

[2]Notation such as "$S \sim \tilde{\Omega}$" indicates that the random set $S$ is drawn from the distribution $\tilde{\Omega}$.

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
