[Reviews · NeurIPS 2018]

Reviewer 1



SUMMARY: The paper studies stochastic versions of classical center-based clustering procedures (e.g. k-center, k-supplier, k-means etc). Such algorithms end up searching for k centers that minimize some objective depending on R_y for each datapoint y where R_y is the distance from y to the closest of the k centers. The stochastic modification allows a distribution over possible clusterings (i.e. distribution over sets of k centers) thus giving a randomized solution from some optimized distribution rather than a fixed solution. The approximation guarantees for the classical clustering procedures are typically stated in terms of the worst-case sub-optimality amongst the R_y's. For example, for k-centers, we get for R_y <= 2 T for all y where T is the optimal solution. This paper gives stochastic clustering procedures which simulatenously attain these classical guarantees but also gives a per-point guarantee in expectation which are tighter (i.e. E[R_y] <= 1.592 T for stochastic k-centers). This per-point guarantee forms the basis for the notion of fairness as the idea is to guarantee that in expectation, each datapoint's distance to its closest center is bounded while simulatenously with probability 1, the original (but weaker) bound still holds. The paper then presents the following results: 1. Chance k-covering guarantee: given demand parameters p_y, r_y, the goal is to find a clustering that satisfies P[R_y <= r_y] >= p_y. The paper proposes a procedure which attains P[R_y <= r_y] >= (1 - 1/e) p_y. 2. The paper then gives a iterative randomized rounding based procedure that attains a guarantee of P[R_y <= 9 r_y] >= p_y 3. Then the paper shows that if p_y are all equal or r_y are all equal, then we obtain better approximation guarantees. 4. Then the paper tackles a special case of the k-covering problem which is essentially better known as k-centers or k-supplier depending on the setting. They provide guarantees of form E[R_y] <= c * r_y where c depends on the setting. The paper address the general case, the SCC setting (where the cluster centers is a subset of the datapoints), and the homogenous SCC setting (where it is SCC and the r_y's are all equal). 5. The paper then provides lower bounds to complement the previous guarantees. 6. The paper then gives approximation algorithms for finding distributions of clusterings that aim at satisfying conditions of form E[R_y] <= t_y where t_y is any feasible vector. They then give a procedure with approximation guarantees for this problem with sparsity guarantees using a multiplicative-weights based procedure and then a shrinking procedure which reduces it down to a linear size. 7. At last, the paper studies whether a stochastic clustering can be used to obtain a deterministic clustering with some positive results and lower bounds. COMMENTS: 1. I am not entirely convinced of the fairness angle of this paper. While it is true that we get tighter per-datapoint guarantees, we already had some (weaker) per-datapoint guarantees. In my opinion, this result should be presented as such: an improved approximation guarantee by allowing stochasticity rather than a fairness guarantee. My impression of standard notions of fairness are based on so-called protected groups where we give guarantees of equality/parity amongst the protected groups. Here, we are giving guarantees about each individual. I guess there are also individual notions of fairness but here, each individual is treated the same regardless of feature values. 2. The iterative-randomized rounding technique is interesting, has such a similar technique been observed before? OVERALL: To my knowledge, this paper provides many new ideas and contributions and is a considerable step forward in our understanding of approximation guarantees for these classical clustering procedures. This is very exciting work; well done!

Reviewer 2



The authors approach the problem of fairness in clustering through stochastic bounds on how far each individual data point can be from its associated cluster center. Section 2 defines the problem of chance k-coverage, which mandates that the k-lottery much satify probabilistics bounds for each point. The primary method of solving the probabilistic chance k-coverage uses a LP relaxation approximation framework with iterative rounding. In Section 3, the authors outline the deterministic case and derive constant c such that the expected distance from cluster center is at most c times r_y using a similar rounding algorithm. Lastly the homogenous SCC setting (all upper bounds are equal) leads to a partial cluster algorithm. I enjoyed this paper and its approach to fairness in clustering. Although not my direct area of research, the bounds and reasoning are explained well and thoroughly. I had hoped for a more extensive related works section instead of the more general "fairness is important" introduction. The relevance of clustering to fairness is obvious in the sense that clustering is a common ML and data science technique, but more grounding would help understand which problems in clustering might be more important. The constants derived in the paper for algorithms seem quite large; do we anticipate this will be practically helpful for fairness applications? Modifying the structure of the paper to include a more detailed intro and a brief conclusion would help with better explaining the motivation of the paper as well.

Reviewer 3



This is a theoretical treatment of the following clustering problem: every data point is annotated with a radius r and a probability p, and we are looking to find a solution such that for all points there is a center within r with probability at least p. The authors formulate the problem, and give some LP-rounding based algorithms to give approximate solutions (relaxing either r or p). First, if the authors want to frame this problem as a 'fair' clustering version, they should do a more thorough review of both the fairness literature, and the emerging fair clustering literature (see, e.g. ICML, NIPS 2017), and place their result among the related work. From a problem statement side, It is not clear at all where the values p and r come from, and who sets them. On the other hand, I view this as an interesting combinatorial problem, with more of a selfish/game theoretic flavor, with the "center" balancing the needs of the individual participants through randomization. In that light, this is a clean and natural problem. However, the theoretical treatment provided by the authors is convoluted and not clear. Any empirical analysis is missing, my guess is due to the impracticality of the LP. On the theory side, first, I would recommend that the authors stick to standard notation -- I do not believe R_{y, S) is cleaner than the traditional d(y, S). Second, a lot of the subroutines are lacking any intuitive definitions / explanations, and the paper overall is lacking a plan describing how the problem is going to be attacked (e.g. what is accomplished by the LP, by the rounding, etc.), what is the structural difference between different versions of the problem, and so on. Individual questions: In Algorithm 1, the index i never appears inside the loop (line 4) is that by design? In the LP, what is the meaning of b(\calF [condition B2], I assume sum_{i \in calF} b_i ? Is Corollary 2.3 a follow up to Theorem 2.2? I am not sure what it is a follow up to, or if it is a stand alone Lemma. Overall, I appreciate the difficulty of fitting the results into the space provided, but I think the paper requires a thorough rewrite before it is ready for publication.